# Primary Axillary Actinomycosis: A Case Report on the Integration of Culture and Molecular Diagnostics for Accurate Diagnosis of Polymicrobial Infections

**DOI:** 10.3390/microorganisms13030671

**Published:** 2025-03-17

**Authors:** Junko Tezuka, Noriyuki Abe, Hiroshi Tanabe

**Affiliations:** 1Department of Dermatology, Tenri Hospital, Tenri 632-8552, Japan; ds111044@yahoo.co.jp; 2Department of Laboratory Medicine, Tenri Hospital, Tenri 632-8552, Japan; abepenem@tenriyorozu.jp

**Keywords:** Polymicrobial infections, Axillary actinomycosis, *Actinomyces gerencseriae*, sulfur granules, *16S rRNA* sequencing, FFPE-PCR, DNA extraction from paraffin-embedded samples

## Abstract

Actinomycosis is a chronic suppurative granulomatous disease caused by *Actinomyces* spp. Although cutaneous actinomycosis is rare, dermatologists must consider it due to its potential coexistence with other pathogens, often as part of polymicrobial infections. We present a rare case of primary axillary cutaneous actinomycosis in a young woman, likely triggered by cosmetic axillary hair removal and home shaving practices. Histological examination revealed characteristics of actinomycosis, including sulfur granules and Gram-positive filamentous structures. Bacterial cultures failed to isolate *Actinomyces*, but identified *Staphylococcus epidermidis*, *S. aureus* (MRSA), and *Corynebacterium simulans*, suggesting a polymicrobial infection contributing to the inflammatory response. Molecular analysis of DNA extracted from formalin-fixed paraffin-embedded (FFPE) tissue yielded a 675 bp PCR product using *Actinomyces*-specific primers. BLAST analysis confirmed the species as *A. gerencseriae*, establishing the diagnosis of actinomycosis. However, a 1000 bp PCR product obtained using universal 16S rDNA primers could not be sequenced successfully, likely due to the presence of multiple bacterial species. Notably, *Actinomyces* was detected only through molecular methods, while bacterial cultures identified the aforementioned bacteria. This discrepancy between FFPE-PCR results and bacterial culture findings demonstrates a key challenge in the microbiological diagnosis of polymicrobial infections. This case highlights the importance of integrating histopathological, microbiological, and molecular techniques for accurate pathogen identification in polymicrobial infections. The failure to detect *Actinomyces* in bacterial cultures, despite its presence in FFPE-PCR, suggests that conventional culture methods alone may be insufficient for diagnosing such infections. Extended culture durations, selective anaerobic culture techniques, and molecular diagnostic methods are essential for a comprehensive evaluation. Recognizing *Actinomyces* as more than a contaminant is important for timely diagnosis and effective treatment. Increased awareness of its potential involvement in polymicrobial infection should improve clinical outcomes.

## 1. Introduction

*Actinomyces* is a genus of filamentous, Gram-positive, facultatively anaerobic bacteria classified as prokaryotes. They exhibit a radial arrangement resembling fungal hyphae and produce asexual reproductive spores characteristic of fungi [1]. They are anaerobic and part of the normal flora of human mucous membranes in the oral cavity, gastrointestinal tract, and reproductive organs, where they typically exhibit low pathogenicity [2]. However, when mucosal or skin barriers are disrupted by trauma, surgery, or the presence of foreign objects such as intrauterine devices (IUDs), the bacteria may invade tissues, leading to actinomycosis, an endogenous infection [2]. Actinomycosis primarily affects the maxillofacial, gastrointestinal, pulmonary, and female genital regions, with an estimated incidence of 1 per 300,000 in Western populations [3]. Cutaneous actinomycosis is particularly rare, and no prior cases of primary axillary cutaneous actinomycosis have been reported.

Actinomycosis progresses from subacute to chronic phases, typically presenting as a chronic suppurative granulomatous condition. It may arise from deep tissue infections or spread hematogenous from other actinomycotic lesions [3]. Clinically, it causes abscesses, asymptomatic swelling, fibrosis, and fistulae that discharge sulfur granules, a hallmark diagnostic feature [4]. Diagnosis primarily relies on histopathology, with sulfur granules and Gram-positive filamentous structures observed in approximately 75% of cases [2]. The most commonly implicated species, *A. israelii* and *A. gerencseriae*, account for approximately 70% of reported cases [5].

Bacterial culture remains the gold standard for pathogen identification, but *Actinomyces* species grow slowly, require prolonged incubation, and have strict anaerobic conditions, leading to culture positivity rates below 50% [1]. Polymicrobial infections further complicate isolation, as competing bacteria may outcompete *Actinomyces* inhibiting its colony formation. Molecular techniques, including *16S rRNA* gene PCR using universal primers and sequencing, are increasingly used for species identification [6]. Matrix-assisted laser desorption/ionization time-of-flight mass spectrometry (MALDI-TOF MS) has also emerged as a complementary method, although its accuracy depends on database completeness and sample preparation protocols [7]. While these methods typically require cultured bacterial colonies, molecular identification can also be performed on DNA from formalin-fixed paraffin-embedded (FFPE) tissue, enabling retrospective diagnosis when cultures fail.

### 1.1. Actinomyces and Polymicrobial Infections

Recent studies highlight the role of *Actinomyces* in polymicrobial infections, where they interact with other bacteria to exacerbate tissue damage and inflammation. *Actinomyces* can promote lesion formation by facilitating the secretion of tissue-degrading enzymes from co-infecting bacteria, like *Fusobacterium* and *Bacteroides* spp. Polymicrobial mechanisms, including quorum sensing, interbacterial signaling, and biofilm formation, enable chronic infections resistant to host immunity and antibiotics. Könönen et al. described *Actinomyces* interactions across various diseases, emphasizing their role in mixed infections [8]. In dermatology, *Actinomyces* is a co-pathogen in polymicrobial infections such as maxillofacial soft tissue infections, perineal infections, Fournier’s gangrene, and diabetic gangrene [8].

### 1.2. Case Significance

This rare case of primary axillary cutaneous actinomycosis in a healthy young woman demonstrates ambiguity in molecular and culture-based identification. Histopathology confirmed sulfur granules and Gram-positive actinomycotic structures, while FFPE-PCR identified *Actinomyces gerencseriae* (serotype II of *A. israelii*). However, cultures failed to isolate *Actinomyces*, instead detecting *Staphylococcus epidermidis*, *S. aureus* (MRSA), and *Corynebacterium simulans*, suggesting a polymicrobial infection. Failure to sequence the *16S rRNA* PCR amplicon suggests the presence of multiple bacteria. The findings reveal a key limitation of culture-based diagnostics, which is their inability to reliably detect anaerobic bacteria in polymicrobial infections, illustrating the importance of molecular methods in such cases.

### 1.3. Recognizing Polymicrobial Infections in Actinomycosis

This case emphasizes the need to integrate histopathological, microbiological, and molecular diagnostics for accurate identification and management of rare *Actinomyces* soft tissue infections. Discordance between FFPE-PCR and culture findings suggest polymicrobial infections when interpreting microbiological results. *Actinomyces* as more than a contaminant is essential for timely diagnosis and appropriate antimicrobial therapy. This case is a valuable reference for diagnostic strategies and clinical management of cutaneous polymicrobial infections involving *Actinomyces*.

## 2. Materials and Methods

### 2.1. Dermoscopy

Dermatoscopic examination was performed using a Dermo Camera (model DZ-D100, Casio Computer Co., Ltd., Tokyo, Japan) to visualize and document the lesion, enabling detailed observation of subtle skin changes not visible to the naked eye [9].

### 2.2. Ultrasound Imaging

Ultrasound imaging was conducted with the LOGIQ E10 system (GE Healthcare, Chicago, IL, USA) using a 15 MHz linear probe. Color Doppler mode assessed blood flow within and around the mass.

### 2.3. Skin Biopsy

The skin biopsy was obtained from the subcutaneous mass in the left axilla, extending from an adjacent erythematous papule, following standard protocols. The site was disinfected with 0.02% benzalkonium chloride, and local anesthesia was administered using 0.5% lidocaine with epinephrine. A 4 mm punch biopsy was performed, and the wound was closed with 4–0 nylon sutures using a simple interrupted technique. Both the excised tissue and the white stony mass from the biopsy site were fixed in 10% neutral buffered formalin for histopathological examination.

### 2.4. Microbial Culture Examination

Discharge from the axillary biopsy site was collected using a sterile swab for bacterial culture. Both aerobic and anaerobic cultures were performed following standard protocols [10].

Aerobic cultures: Blood agar plates, incubated at 35 °C for 48 h.Anaerobic cultures: Brucella HK agar plates, incubated at 35 °C for 48 h.

Aerobic cultures were identified using matrix-assisted laser desorption/ionization-time of flight mass spectrometry (MALDI-TOF MS) (MALDI Biotyper^®^ version 3.1, Bruker Daltonics, Billerica, MA, USA) [11]. *Actinomyces* species were not isolated, as specific cultures were not performed.

### 2.5. Mass Spectrometry Analysis

Axillary skin biopsy samples were collected using sterile cotton swabs.

Sample preparation: Bacterial extracts (1 μL) obtained with 70% formic acid were spotted onto a MALDI target plate (MSP 96 target, polished steel, Bruker Daltonics, Billerica, MA, USA), air-dried and overlaid with 1 μL of matrix solution HCCA (α-4-cyano-hydroxycinnamic acid in 50% acetonitrile and 2.5% trifluoroacetic acid).

Mass spectrometry: Spectral profiles were acquired using a Microflex (Bruker Daltonics, Germany) in linear positive ion mode optimized for the m/z range of 2000–20,000. Data processing: Spectral data were analyzed using flexControl 2.4, MALDI Biotyper 3.0, and flex Analysis 2.4, Bruker Daltonics, Billerica, MA, USA.

Identification criteria: Spectra were compared with the MALDI Biotyper 3.0 library (version 3.2.1.1) using a composite correlation index (CCI) matrix:Score ≥ 2.0: Species-level identification.Score 1.7–2.0: Genus-level identification [12].

### 2.6. Histopathological Examination

Formalin-fixed tissue, including the mass from the biopsy site, was examined using three staining methods:Hematoxylin and Eosin (H&E) Staining: Evaluated histological features and tissue structures [13,14].Gram Staining: Differentiated Gram-positive and Gram-negative bacteria. *Actinomyces* stain violet confirming its Gram-positive nature [15].Ziehl-Neelsen Staining (Fite Method): Screened for acid-fast bacilli, excluding atypical mycobacteria. No acid-fast organisms were detected [16].

### 2.7. Molecular Biological Analysis

Molecular identification was conducted on FFPE biopsy specimens. Bacterial DNA was extracted from five 10-µm FFPE tissue sections using the QIAamp DNA FFPE Tissue Kit (Qiagen, Tokyo, Japan; catalog no. 56404) [17,18]

#### 2.7.1. PCR for 16S rRNA Gene

To identify the bacterial species observed in culture within the FFPE biopsy specimens, we performed PCR amplification targeting a conserved region of the bacterial *16S rRNA* gene, facilitating subsequent sequencing and precise identification.

Bacterial DNA was extracted from five 10-µm FFPE tissue sections using the QIAamp DNA FFPE Tissue Kit (Qiagen, Tokyo, Japan; catalog no. 56404). The extracted DNA served as the template for PCR amplification.

The PCR reaction mixture (total volume of 20 µL) was prepared as follows:Master Mix: 10 µL of GoTaq^®^ Master Mix (Promega, Madison, WI, USA; catalog no. M7122), which includes Taq DNA polymerase, dNTPs, and reaction buffers.Primers: 0.5 µM each of universal bacterial primers:Forward (8UA primer): 5′-AGAGTTTGATCMTGGCTCAG-3′Reverse (1485B primer): 5′-TACGGYTACCTTGTTACGACTT-3′Template DNA: 2 µL of extracted bacterial DNA.

The PCR cycling conditions were set as follows:


Initial denaturation: 94 °C for 7 min to denature the DNA strands.Amplification cycles (35 cycles):○Denaturation: 95 °C for 20 s.○Annealing: 60 °C for 20 s to allow primer binding.○Extension: 72 °C for 40 s for DNA strand elongation.○Final extension: 72 °C for 7 min to ensure complete extension of all PCR products.


Post-amplification, the PCR products were subjected to electrophoresis on 2% agarose gels containing ethidium bromide (0.5 µg/mL) in TAE buffer. Electrophoresis was conducted at 100 V for 30–45 min. The gels were then visualized under UV light to confirm the presence and size of the amplified fragments, ensuring the specificity of the PCR amplification [19].

#### 2.7.2. PCR for Actinomyces-Specific Identification

To specifically detect *Actinomyces* species in the FFPE biopsy specimens, we performed PCR amplification targeting the *Actinomyces*-specific gene. The reaction aimed to amplify a conserved region unique to *Actinomyces*, facilitating subsequent sequencing and precise identification.

##### DNA Extraction

Bacterial DNA was extracted from five 10-µm FFPE tissue sections using the QIAamp DNA FFPE Tissue Kit (Qiagen, Tokyo, Japan; catalog no. 56404). The extracted DNA served as the template for PCR amplification.

##### PCR Reaction Mixture (Total Volume of 20 µL)

Master Mix: 10 µL of GoTaq^®^ Master Mix (Promega; catalog no. M7122), which includes Taq DNA polymerase, dNTPs, and reaction buffers.Primers: 0.5 µM each of *Actinomyces*-specific primers:Forward: 5′-GGCTTGCGGTGGTACGGGC-3′Reverse: 5’-GGCTTTAAGGGATTGCGTCCACCTCAC-3′Template DNA: 2 µL of extracted bacterial DNA.

##### PCR Cycling Conditions

Initial denaturation: 94 °C for 7 min to denature the DNA strands.Amplification cycles (35 cycles):○Denaturation: 95 °C for 20 s.○Annealing: 60 °C for 20 s to allow primer binding.○Extension: 72 °C for 40 s for DNA strand elongation.Final extension: 72 °C for 7 min to ensure complete extension of all PCR products.

##### Post-Amplification Analysis

○A 675 bp PCR product was obtained, indicative of *Actinomyces* presence. The specificity of the amplification was confirmed through electrophoresis and subsequent sequencing [20].

#### 2.7.3. DNA Sequencing and Analysis

PCR products were purified using the QIAquick Gel Extraction Kit (Qiagen, Hilden, Germany; catalog no. 28704), following the manufacturer’s instructions. Sequencing was performed with ABI PRISM^®^ BigDye™ Terminator Cycle Sequencing Kits (Applied Biosystems, Foster City, CA, USA) on an ABI310 DNA Genetic Analyzer (Applied Biosystems). Sequences were analyzed using GENETYX^®^ software (Genetyx Ver.13.0, Tokyo, Japan) and compared to public *16S rRNA* gene databases for bacterial identification.

## 3. Case Report

### 3.1. Clinical Presentation

A 24-year-old healthy woman presented with a subcutaneous mass in her left axilla, first noticed a year earlier. Initially asymptomatic, the mass gradually enlarged, developing a red papule on its surface, prompting medical consultation.

At the initial examination, a firm subcutaneous mass with a smooth-surfaced red papule resembling granulation tissue was observed (Figure 1). Dermoscopy and ultrasound imaging revealed a 16-mm hypoechoic mass with internal vascularization, contiguous with the surface papule (Figure 2).

The patient had no underlying health conditions, and her laboratory data were within normal range. She had no history of chronic diseases such as diabetes, obesity, or immunosuppressive conditions, including corticosteroid therapy or malignancies.

### 3.2. Histopathological Findings

A skin biopsy of the papule revealed a 4 mm white stony mass, submitted for pathological examination. Pus from the lesion was also collected for bacterial culture. H&E staining showed supprative granulation tissue with dense infiltrates of neutrophils and plasma cell infiltrates, hemorrhage, and vascular dilatation in the dermis (Figure 3). The stony mass exhibited basophilic filamentous structures surrounded by eosinophilic clubs and inflammatory cells, predominantly polymorphonuclear neutrophils. Gram staining confirmed densely Gram-positive branching structures, consistent with sulfur granules, a hallmark of actinomycosis (Figure 4). Ziehl-Neelsen staining was negative, ruling out *Nocardia* species.

### 3.3. Bacterial Culture and MALDI-TOF MS Identification

Pus culture revealed the following organisms:*S. epidermidis* (coagulase-negative staphylococcus, CNS, 1+)*S. aureus* (methicillin-resistant *S. aureus,* MRSA, 1+)*C. simulans* (1+)

Species identification was performed using MALDI-TOF MS (MALDI Biotyper^®^ version 3.1, Bruker Daltonics, Billerica, MA, USA). Due to low bacterial counts, the clinical significance of CNS and MRSA remained unclear. No anaerobic bacteria were detected, and *Actinomyces* species were not isolated in culture. Therefore, MALDI-TOF MS analysis for *Actinomyces* was not performed.

### 3.4. Molecular Identification of Actinomyces

DNA was extracted from FFPE tissue using the QIAamp DNA FFPE Tissue Kit (Qiagen, Tokyo, Japan; catalog no. 56404) based on histopathological suspicion of actinomycosis. PCR targeting the *16S rRNA* gene amplified a 675-bp fragment specific to *Actinomyces*. Gel electrophoresis confirmed the presence of distinct bands, including the 675-bp amplicon. The sequence was analyzed via BLAST (NCBI, accessed on March 13, 2025), identifying the pathogen as *A. gerencseriae* (formerly *A. israelii* serotype II) with 99.37% sequence identity and 97% query coverage to strain KCOM 3578 (GenBank accession: MW541884.1) [21]. Sequence data obtained from the FFPE sample are presented in Appendix A. Additionally, a 1000-bp universal bacterial *16S rRNA* fragment was amplified, but BLAST analysis found no definitive species match, suggesting the presence of multiple bacterial species within the sample (Figure 5).

Conventional bacterial culture failed to isolate *Actinomyces* and instead identified non-*Actinomyces* bacteria that were not detected through FFPE-PCR. This highlights the limitations of culture-based methods in polymicrobial infections and emphasizes the complementary roles of both molecular identification and culture techniques in diagnosing such infections.

### 3.5. Clinical Diagnosis

Since histopathological findings revealed chronic granulomatous inflammation with sulfur granules, consistent with actinomycosis, and also that Gram staining identified filamentous Gram-positive bacteria, cutaneous actinomycosis was suspected. Computed tomography imaging of the neck to pelvis showed no systemic dissemination, confirming primary axillary cutaneous actinomycosis.

### 3.6. Treatment and Clinical Outcome

The patient was treated with oral amoxicillin hydrate (250 mg, six tablets/day) for 2-months, leading to complete resolution of the mass (Figure 6). No recurrence was observed at 6-months after cessation of treatment.

## 4. Discussion

### 4.1. Clinical Features and Diagnosis

This case involves cutaneous actinomycosis in the axilla of a woman in her 20s, a rare presentation suspected to have been triggered by hair shaving with a contaminated razor. The diagnosis was confirmed by the presence of sulfur granules and Gram-positive filamentous bacteria. Although the causative species was not detected via culture, molecular analysis of FFPE tissue from the affected site identified *A. gerencseriae*. This discrepancy between culture and molecular findings highlights the need for further discussion on the emerging role of *Actinomyces* spp. in polymicrobial infections.

### 4.2. Clinical Implications

This case highlights several key points:1.Advanced Diagnostics

Clinicians should extend culture durations and use molecular identification and mass spectrometry to detect slow-growing organisms or those difficult to culture like *Actinomyces* spp.

2.Polymicrobial Context

The role of *Actinomyces* and co-infecting bacteria in skin and soft tissue infections should not be overlooked. In polymicrobial infections, optimizing anaerobic culture conditions and extending culture durations are essential to avoid misclassification of *Actinomyces* as contaminants.

3.Recognition of Rare Presentations

Primary axillary cutaneous actinomycosis is extremely rare. Recognizing unusual clinical presentations and accurately identifying causative pathogens can improve understanding of disease mechanisms, inform treatment decisions, and guide patient education, including proper shaving techniques.

### 4.3. Discrepancy Between Culture and Molecular Identification

A key issue in this case is the mismatch between bacterial species identified through culture and those detected via molecular techniques. While culture revealed *S. epidermidis*, *S. aureus* (MRSA), and *C. simulans*, molecular analysis identified only *A. gerencseriae*. Several factors may explain this discrepancy:1.Bacterial Abundance and Sampling Bias

PCR preferentially amplifies DNA from organisms present in higher abundance or with structurally intact DNA. In this case, *Actinomyces* DNA from sulfur granules may have predominated, suggesting it as the primary pathogen.

2.Challenges in Culturing *Actinomyces* spp. [22]

Some *Actinomyces* spp. are microaerophilic or anaerobic, making them difficult to isolate under standard culture conditions. Additionally, short incubation periods may lead to misclassification as contaminants. In contrast, PCR can detect these organisms regardless of growth conditions, highlighting the limitations of culture-based diagnostics.

3.PCR Amplification Bias [23]

Molecular techniques favor high-quality or easily amplifiable DNA, potentially introducing detection biases. DNA in FFPE blocks may be fragmented or damaged, reducing amplification efficiency. Primer design and sequence complexity can also influence bacterial detection rates.

4.Differences in Tissue and Pus Composition [24,25]

Pus may contain secondary contaminants, whereas sulfur granules more accurately represent the primary infection site. Additionally, inhibitors introduced by paraffin embedding can affect DNA extraction and PCR performance.

These findings highlight the complementary roles of culture-based and molecular diagnostics. While culture tests can identify viable organisms, molecular techniques can detect unculturable pathogens, offering more information in cases of polymicrobial infections.

### 4.4. Role of Actinomyces *spp.* in Polymicrobial Infections

Recent studies highlight the role of *Actinomyces* spp. in polymicrobial infections. In most *Actinomyces* -associated infections, co-infecting bacteria such as *Streptococcus* spp., *Staphylococcus* spp., *Corynebacterium* spp., and *Fusobacterium* spp. are also present [26]. These organisms interact synergistically with *Actinomyces,* impairing host defenses and reducing oxygen tension in infected tissues, thereby promoting *Actinomyces* growth [27]. Thus, *Actinomyces* spp. not only function as standalone pathogens but also as contributors to polymicrobial infections. Determining and addressing these interactions may improve outcomes in the management of actinomycosis. In this case, co-infecting bacteria likely exacerbated the inflammatory response and clinical symptoms, complicating disease progression. These findings reinforce the need to consider *Actinomyces* spp. as potential co-pathogens in skin and soft tissue infections.

### 4.5. Treatment and Outcomes

Traditionally, actinomycosis is treated with prolonged high-dose penicillin G or amoxicillin for 6–12 months. However, in this case, complete resolution was achieved with a 2-month course of oral amoxicillin and appropriate drainage, with no recurrence over 6-months. This shorter treatment duration suggests that early diagnosis and timely intervention can reduce the need for prolonged antibiotic therapy, minimizing patient burden.

## 5. Conclusions

This case demonstrates the efficacy of an interdisciplinary approach, integrating histopathology, microbiology, and molecular biology. It also illustrates the expanding pathogenic role of *Actinomyces* spp. in polymicrobial infections, reinforcing the need for heightened awareness and tailored treatment strategies for effective management.

## Figures and Tables

**Figure 1 microorganisms-13-00671-f001:**
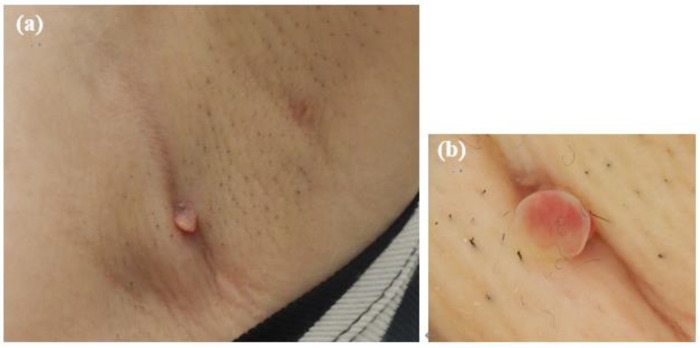
Initial presentation of the axillary lesion. (**a**) A subcutaneous mass and a red papule in the left axilla. (**b**) Dermoscopic image of the papule.

**Figure 2 microorganisms-13-00671-f002:**
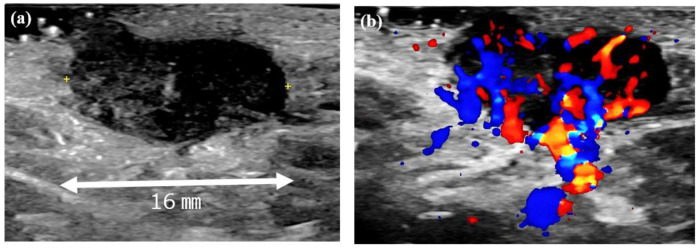
Ultrasound images of the left axillary mass. (**a**) A 16-mm hypoechoic mass with no posterior enhancement. (**b**) Doppler ultrasound image showing abundant internal blood flow and continuity with the surface papule.

**Figure 3 microorganisms-13-00671-f003:**
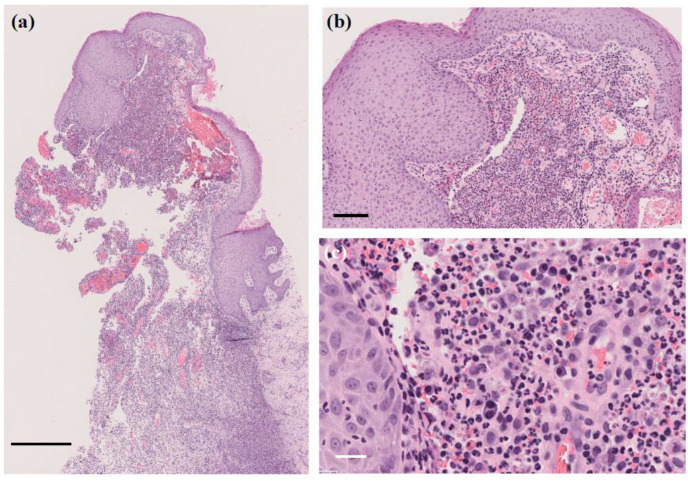
H&E staining of the papule. Suppurative granulation tissue with dense infiltrates of neutrophils and plasma cells, hemorrhage, and vascular dilatation in the dermis. (**a**) Original magnification ×40. Scale bar = 0.5 mm. (**b**) Original magnification ×200. Scale bar = 100 μm. (**c**) Original magnification ×400. Scale bar = 20 μm.

**Figure 4 microorganisms-13-00671-f004:**
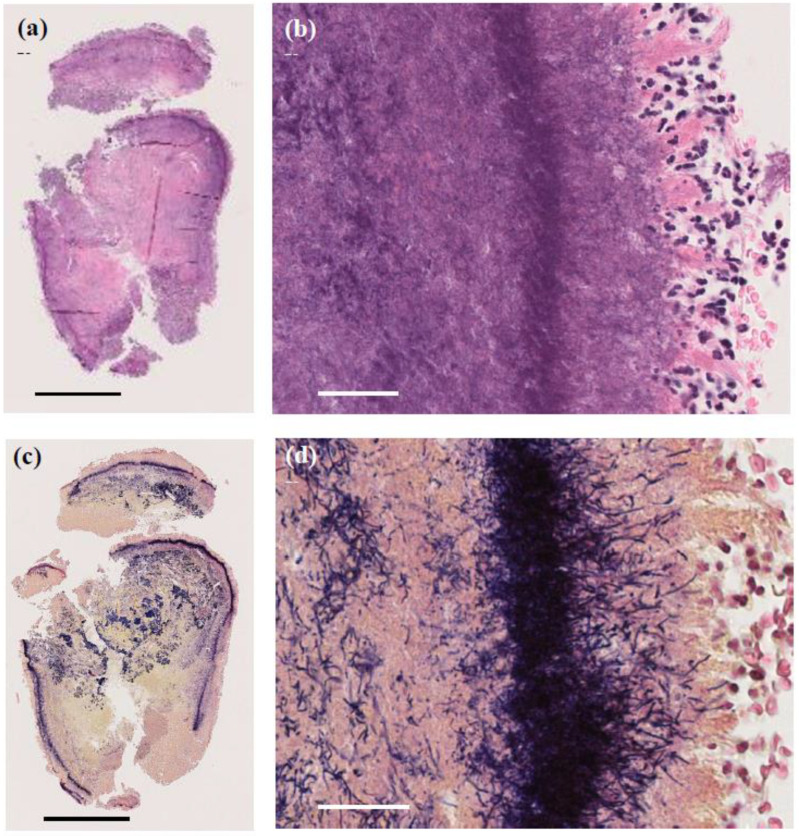
Pathological images of the 4 mm granule discharged from the lesion. (**a**,**b**) H&E staining showing basophilic filamentous structures surrounded by eosinophilic clubs and inflammatory cells, predominantly polymorphonuclear neutrophils. (**a**) Original magnification ×40. Scale bar = 1.0 mm. (**b**) Original magnification ×400. Scale bar = 40 μm. (**c**,**d**) Gram staining revealing Gram-positive filamentous structures, characteristic of sulfur granules in actinomycosis. (**c**) Original magnification ×40. Scale bar = 1.0 mm. (**d**) Original magnification ×400. Scale bar = 40 μm.

**Figure 5 microorganisms-13-00671-f005:**
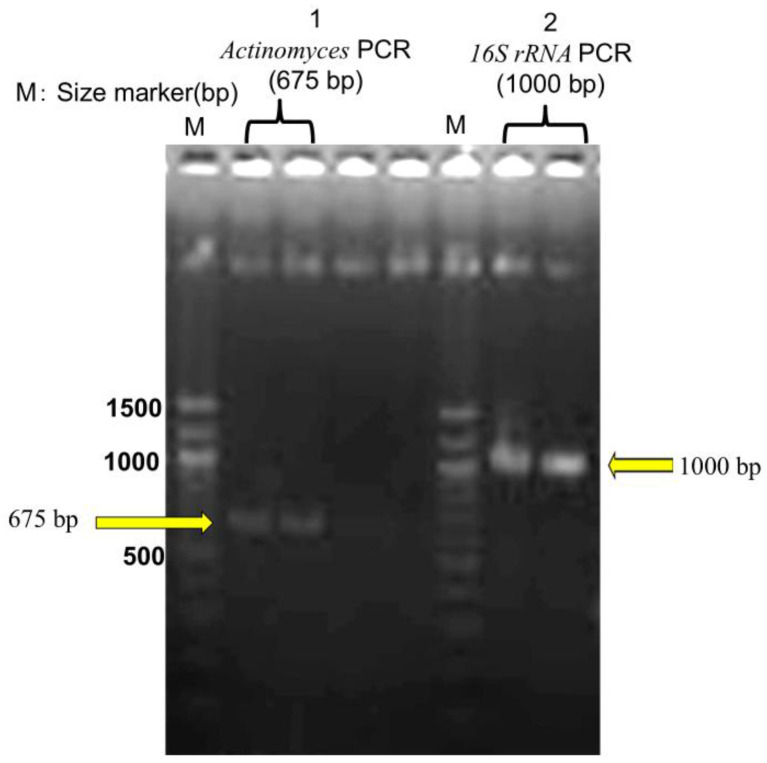
Agarose gel electrophoresis of PCR products. M: DNA size markers. Lane 1: *Actinomyces* PCR product (~675 bp). Lane 2: *16S rRNA* PCR product (~1000 bp). Each sample was analyzed in duplicate to ensure reproducibility.

**Figure 6 microorganisms-13-00671-f006:**
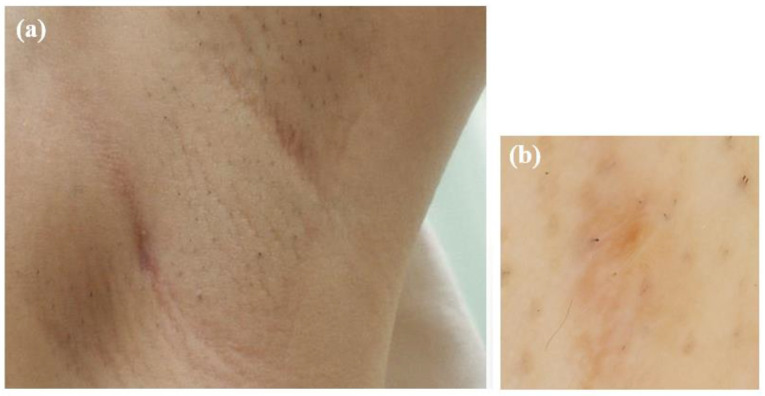
Photographs taken after two months of oral amoxicillin treatment. (**a**) The subcutaneous mass had resolved. (**b**) Dermoscopic image showing the disappearance of the red papule.

## Data Availability

The data that support the findings of this case report are available from the corresponding author upon reasonable request.

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
