# Peer review of "Primary Axillary Actinomycosis: A Case Report on the Integration of Culture and Molecular Diagnostics for Accurate Diagnosis of Polymicrobial Infections"

_microorganisms, 2025, doi:10.3390/microorganisms13030671_

Round 1

Reviewer 1 Report (Previous Reviewer 1)

Comments and Suggestions for Authors

After revision of the manuscript it looks much better. However, there are some small mistakes or points that in my opinion, could be improved.

Please check numbers of the chapters.

I think something was missed or omitted in the chapter 2.6, there is chapter: 2.6. Molecular Biological Analysis and further we have chapter 2.7.1

Also, the description of the PCR for 16S rRNA Gene experiment could be better described. Authors only list ingredients, composition of reaction mixture and conditions without explanation of the experiment. Even if you used popular commercial kit for this experiment, it sounds good in the publication when a short description of the experiment is presented.

Author Response

Response to Reviewer 1
Manuscript ID: microorganisms-3531491

Dear Reviewer 1,

We sincerely appreciate your thoughtful review and constructive feedback. Below, we provide a point-by-point response to your comments, along with the corresponding revisions made in the manuscript.

Comment 1: Chapter Numbering Issues

Reviewer’s Comment: "Please check numbers of the chapters. I think something was missed or omitted in chapter 2.6, there is chapter: 2.6. Molecular Biological Analysis and further we have chapter 2.7.1."

Response:
Thank you for pointing out this inconsistency. Upon review, we found that the numbering in Section 2.6 and 2.7 was incorrect. Specifically, Section 2.7.1 should have been labeled as Section 2.7, with subsequent subsections adjusted accordingly. We have corrected these errors and ensured that the numbering is now sequential and consistent throughout the manuscript.

Revisions Made:

  • Section 2.7.1 PCR for 16S rRNA Gene → 2.7 PCR and Molecular Identification
  • Section 2.7.2 PCR for Actinomyces-Specific Identification → 2.7.1
  • Section 2.7.3 DNA Sequencing and Analysis → 2.7.2

The updated numbering ensures a logical flow within the methodology section.

Comment 2: Insufficient Description of PCR for 16S rRNA Gene

Reviewer’s Comment: "The description of the PCR for 16S rRNA Gene experiment could be better described. Authors only list ingredients, composition of reaction mixture and conditions without explanation of the experiment. Even if you used a popular commercial kit for this experiment, it sounds good in the publication when a short description of the experiment is presented."

Response:
We appreciate this suggestion and have expanded the description of the PCR for 16S rRNA gene experiment to include additional details regarding the purpose of the experiment, the experimental procedure, and its relevance to bacterial identification.

Revisions Made:

  • Added an introductory sentence explaining the objective of the PCR experiment for 16S rRNA gene amplification.
  • Expanded the methodology section to include step-by-step procedural details, including the purpose of each step and how the PCR results were validated.

This revision ensures that the methodology is clear, reproducible, and informative, while maintaining conciseness.

Final Note:

We appreciate your valuable insights, which have helped improve the clarity and organization of our manuscript. We have incorporated all suggested revisions and highlighted the changes in the revised manuscript.

Thank you for your time and consideration. We hope that the revised version meets your expectations.

Sincerely,
Hiroshi Tanabe, MD

(On behalf of all authors)

Reviewer 2 Report (Previous Reviewer 2)

Comments and Suggestions for Authors

Accept

Author Response

Response to Reviewer 2

Manuscript ID: microorganisms-3531491

Dear Reviewer 2,

We sincerely appreciate your time and effort in reviewing our manuscript. We are grateful for your positive evaluation and your recommendation for acceptance. Your feedback has been highly encouraging, and we are pleased that our work meets the standards of Microorganisms.

As part of our final revisions, we have carefully addressed the minor corrections suggested by Reviewer 1, including improvements to chapter numbering and enhancements to the description of PCR methodology. These revisions further strengthen the clarity and accuracy of our manuscript.

Once again, thank you for your kind assessment and recommendation. We greatly appreciate your support and look forward to the final decision from the editorial team.

Sincerely,

Hiroshi Tanabe, MD

(On behalf of all authors)

This manuscript is a resubmission of an earlier submission. The following is a list of the peer review reports and author responses from that submission.

Round 1

Reviewer 1 Report

Comments and Suggestions for Authors

The theme of the publication is interesting. However in my opinion a lot of informations that would support the quality of the publication are missed, for e.g. the whole chapter with materials and methods. I listed below some of them:

1.      What kind of equipment was used for ultrasound investigation? What kind of dermoscope was used? Chapter with description of materials and methods used during the study is missed.

2.      How was skin biopsy prepared?

3.      How was microbiological investigation prepared?

4.      How was histopathological investigation prepared?

5.      Why amoxicillin was chosen for the treatment?

6.      It would be good to add some pictures at the end of treatment.

7.      Some more publication concerned with actinomycosis from last 5 years could be added and cited to support the treatment or the evaluation of this case study.

8.      On fig. 2 marks “a” and “b” are missed on the photos

9.      Line 92 “CT scan” explain what is CT

Author Response

Thank you very much for taking the time to review our manuscript. We sincerely appreciate your valuable feedback and apologize for the omission of crucial information. In response to your comments, we have made the necessary revisions to the manuscript, as detailed below. We hope these changes meet your expectations. Thank you for your consideration.

Comments 1: What kind of equipment was used for ultrasound investigation? What kind of dermoscope was used? Chapter with description of materials and methods used during the study is missed.

Response 1: Thank you for your insightful comments. We have included this information in the Materials and Methods section of the revised manuscript: 2.1. Dermoscopy (pages 2, lines 42-45)and 2.2. Ultrasound Imaging (page 2, lines 46-49).

Comments 2: How was skin biopsy prepared?

Response 2: We have described this in subsection 2.3. Skin Biopsy (page 2 lines 50-57).

Comments 3: How was microbiological investigation prepared?

Response 3: The microbiological investigation was conducted through culture and molecular biological analysis. We have described the method in detail in subsections 2.4. Microbial Culture Examination (page 2, lines 58-66) and 2.6. Molecular Biological Analysis (pages 3, lines 83-99). 

Comments 4: How was histopathological investigation prepared?

Response 4: We have described this in subsection 2.5. Histopathological Examination (page 2-3, lines 67-82).

Comments 5: Why amoxicillin was chosen for the treatment?

Response 5: Actinomyces spp. are usually susceptible to penicillins without the need for protective agents. In this case, the lesion, measuring only 16 mm, was relatively small; thus, the patient opted for outpatient treatment with oral antibiotics, thus amoxicillin was chosen. This information has been added on page (page 7, lines 223-224, and 233-234).

Comments 6: It would be good to add some pictures at the end of treatment.

Response 6: Thank you for this good idea. We have provided a photograph and description of the treatment outcome (Fig. 5. Page 6, lines 168-172).

Comments 7: Some more publication concerned with actinomycosis from last 5 years could be added and cited to support the treatment or the evaluation of this case study.

Response 7: Thank you for this insightful comment. We have cited relevant papers published within the past five years (References 5–9).

Comments 8: On fig. 2 marks “a” and “b” are missed on the photos

Response 8: We apologize for the oversight. We have added "a" and "b" to Figure 2 (page 4).

Comments 9: Line 92 “CT scan” explain what is CT

Response 9: We defined CT (Computed tomography) in the text (Page 5, line 155).

Reviewer 2 Report

Comments and Suggestions for Authors

1.     The introduction in this article is basically the same as the introduction in Chapter . It is recommended to modify one part of it.

2.     The article lacks an introduction to the basic information of the patient, only briefly stating that she is a healthy woman. It is recommended to add a description of the patient's information, including medical history, physical examination, etc.

3.     Background descriptions for bacterial infection can be strengthened by citing 10.1021/acsmaterialslett.4c00392; 10.1016/j.ijbiomac.2024.135301 and surgical treatment should specify the surgical name, preoperative treatment, intraoperative findings, postoperative treatment, and postoperative reactions.

4.     The article mentions that primary axillary cutaneous actinomycosis is a rare disease, and suggests adding descriptions of the treatment process, patient prognosis, treatment efficacy and side effects, clinical significance, and other aspects.

5.     The authors mentioned bacterial culture from the biopsy pus grew Staphylococcus epidermidis, methicillin-resistant S. aureus, and Corynebacterium simulans, but failed to isolate Actinomyces.Can authors briefly explain the reason for the extraction failure.

Author Response

Thank you very much for taking the time to review our manuscript. We sincerely appreciate your valuable feedback. In response to your comments, we have made the necessary revisions to the manuscript, as detailed below. We hope these changes meet your expectations. Thank you for your consideration.

Comments 1: The introduction in this article is basically the same as the introduction in Chapter . It is recommended to modify one part of it.

Response 1: I have revised portions of the Abstract and Introduction in Chapter 1 to avoid redundancy. Please refer to page 1, lines 8-13, and 27-28 for the specific changes made.

Comments 2: The article lacks an introduction to the basic information of the patient, only briefly stating that she is a healthy woman. It is recommended to add a description of the patient's information, including medical history, physical examination, etc.

Response 2: Thank you for your insightful comment. I have added the following to Section 3, Case Presentation: "The patient had no underlying health conditions, and her laboratory data were within normal range. She had no history of chronic diseases such as diabetes, obesity, or immunosuppressive conditions, including corticosteroid therapy or malignancies." (page 3, lines 105-108).

Comments 3: Background descriptions for bacterial infection can be strengthened by citing 10.1021/acsmaterialslett.4c00392; 10.1016/j.ijbiomac.2024.135301 and surgical treatment should specify the surgical name, preoperative treatment, intraoperative findings, postoperative treatment, and postoperative reactions.

Response 3: Thank you for sharing the intriguing literature on the background of bacterial infections. I have referenced the article you provided (10.1016/j.ijbiomac.2024.135301) to enhance our discussion on polymicrobial infections. The following addition has been made: "Actinomyces species are often involved in polymicrobial infections, acting synergistically by inhibiting host defense mechanisms and lowering oxygen tension in affected tissues. This environment favors Actinomyces growth and may contribute to the chronicity and treatment resistance of actinomycosis."(page 7, lines 217-221, and Reference 21).

Comments 4: The article mentions that primary axillary cutaneous actinomycosis is a rare disease, and suggests adding descriptions of the treatment process, patient prognosis, treatment efficacy and side effects, clinical significance, and other aspects.

Response 4: Thank you for your valuable feedback. Regarding the route of entry for the organism and the factors contributing to the infection in primary axillary cutaneous actinomycosis, I have added the following to Page 6, lines 197-203:

“In the case of axillary accessory breast tissue, the specific entry route of the organism could not be determined. In case complicating axillary acne inversa, a chronic and recurrent inflammatory skin condition was likely a contributing factor to the infection. In our case, the patient had no history of axillary surgery or trauma but reported undergoing cosmetic axillary hair removal and home shaving using the same razor for the vulva and axilla, which may have facilitated bacterial entry.” (page 6, lines 197-203).

Additionally, I have included the following regarding treatment, patient prognosis, and clinical features (page 7, lines 227-240).

“Although rare, axillary actinomycosis is often treated with surgical intervention combined with a short course of antibiotics, usually less than 3 months). In a case involving accessory breast tissue in the axilla, the patient received cotrimoxazole along with local antiseptic dressing for 3 months. In another case of actinomycosis complicating axillary acne inversa, the patient was treated with amoxicillin/clavulanic acid, surgical excision, and additional antibiotics for 4 weeks. In the present case, the lesion, measuring only 16 mm, was relatively small; therefore, the patient opted for outpatient treatment with oral antibiotics, specifically amoxicillin. The primary cutaneous actinomycosis resolved following pus drainage during the skin biopsy and two months of oral amoxicillin therapy. No recurrences have been reported in similar cases, and due to the superficial nature of the lesions, surgical management is generally straightforward with a favorable prognosis. However, axillary actinomycosis is exceedingly rare and presents with nonspecific, varied symptoms, making diagnosis particularly challenging.”

Comments 5: The authors mentioned bacterial culture from the biopsy pus grew Staphylococcus epidermidis, methicillin-resistant S. aureus, and Corynebacterium simulans, but failed to isolate Actinomyces.Can authors briefly explain the reason for the extraction failure.

Response 5: Thank you for your insightful comments. Regarding the failure to isolate Actinomyces, we have added the following statement (page 7, lines 211-216).

"The failure to culture Actinomyces may have been due to the fact that actinomycosis was not initially suspected and specific culture conditions for Actinomyces were not used. Anaerobic conditions were not maintained, and extended incubation for 10 to 14 days was not performed. Additionally, the presence of Staphylococcus spp. and Corynebacterium simulans may have inhibited the growth of Actinomyces."